# Tear matrix metalloproteinase-9 levels may help to follow a ocular surface injury in lagophthalmic eyes

**Marcela Minaříková**[1], **Zdeněk Fík**[2], **Josef Štorm**[3], **Kateřina Helisová**[4], **Květoslava Ferrová**[3], **Gabriela Mahelková**[1,3]*

**1** Department of Physiology, 2nd Faculty of Medicine, Charles University, Prague, Czech Republic, **2** Department of Otorhinolaryngology and Head and Neck Surgery, 1st Faculty of Medicine, Charles University and Motol University Hospital, Prague, Czech Republic, **3** Department of Ophthalmology, 2nd Faculty of Medicine, Charles University and Motol University Hospital, Prague, Czech Republic, **4** Department of Mathematics, Faculty of Electrical Engineering, Czech Technical University in Prague, Prague, Czech Republic

* gabriela.mahelkova@fnmotol.cz

**Data Availability Statement:** All relevant data are within the paper. The minimal dataset underlying our results described in our paper is presented within Tables 1 and 2 of the paper.

## Abstract

The preocular tear film is critically important for maintaining healthy ocular surface. In lagophthalmos, increased evaporation and tear film instability can occur. The level of tear matrix metalloproteinase 9 (MMP-9) is considered as a possible marker of ocular surface damage and inflammation. The aim of this study was to evaluate the possible usefulness of measuring tear film levels of MMP-9 in patients with lagophthalmos. Sixteen adult patients with unilateral lagophthalmos due to cerebellopontine angle mass surgery were included. Basic clinical examination including tear film osmolarity, degree of lagophthalmos, ocular surface sensitivity testing, corneal fluorescein staining, and tear break-up time (TBUT) were performed. Furthermore, tear MMP-9 quantification was performed and the values from lagophthalmic and contralateral healthy eye were compared. Possible correlations between tear MMP-9 levels and other parameters were analyzed. The Oxford score was higher in lagophthalmic eyes in comparison to healthy eyes. TBUT and corneal sensitivity were lower in lagophthalmic eyes. There was no difference in osmolarity between the two groups. Tear MMP-9 values were higher in lagophthalmic eyes. A higher MMP-9 value was associated with an increase in ocular surface fluorescein staining and a decrease of TBUT in lagophthalmic eyes. Tear MMP-9 may be used for monitoring ocular surface damage, contribute to early detection of inflammation progression and facilitate treatment adjustments.

## Introduction

The preocular tear film is critically important in maintaining healthy ocular surface. Tear film homeostasis preservation is a highly complex process controlled by the lacrimal functional unit, eyelids, and interconnecting sensory and motor nerves [1, 2]. Tear film instability may result in tear fluid hyperosmolarity which can stress and potentially alter the underlying ocular surface resulting in inflammation [3, 4].

**Funding:** The project was supported by the project of Second Faculty of Medicine, Charles University, Prague No. 1110026 and by the project (Ministry of Health, Czech Republic) for conceptual development of research organization 00064203 (University Hospital Motol, Prague, Czech Republic).

**Competing interests:** The authors have declared that no competing interests exist.

Full eyelid closure when blinking is necessary to maintain a stable tear film and healthy ocular surface. Patients who are unable to blink or to completely close their eyes are at risk of corneal-related disorders, including dry eye symptoms, corneal exposure, tear film evaporation, and subsequent exposure keratopathy [5].

Lagophthalmos is, in essence, an incomplete or abnormal closure of the eyelids, and it can cause increased evaporation and severe tear film instability. One of the possible causes of lagophthalmos is facial nerve paralysis following cerebellopontine angle tumor surgery. Tear secretion in these patients may be disturbed as well due to lacrimal nerve damage [6]. However, the disability is often transient. The main goal of lagophthalmos treatment is protecting the ocular surface and preserving unaltered vision. Close monitoring of the ocular surface is necessary to prevent complications and helps in determining proper management.

Matrix metalloproteinases are a multidomain calcium and zinc ion-dependent enzyme family and play an important role in many physiological and pathological processes, including tissue remodeling, wound healing, and inflammation. Increased levels of tear matrix metalloproteinase-9 (MMP-9) were described in patients with dry eye disease and other conditions associated with ocular surface damage, and tear MMP-9 was considered a possible marker of ocular surface damage and inflammation [4, 7, 8].

The aim of this study was to assess potential benefits usefulness of measuring tear film MMP-9 levels in patients with lagophthalmos after cerebellopontine angle tumor surgery.

## Materials and methods

This was a cross sectional study of 16 enrolled adult Caucasian patients attending the Ophthalmology Department of Motol University Hospital, with unilateral lagophthalmos following cerebellopontine angle mass surgery (vestibular schwannoma; VS) in 2018–2020. Only patients that used no other local treatment than topical artificial tear drops or ointment treatment were included. Patients using other types of topical therapy within three months of the examination, patients after anterior segment surgery, and contact lens wearers were excluded. Patients with systemic disorders that may affect the ocular surface were excluded. Patients with severe epithelial defects or corneal ulcers were also excluded. Written informed consent was obtained from each patient after explaining the nature and possible consequences of the study. The study was approved by the Institutional Review Board (IRB)–The Ethics Committee of the Motol University Hospital. The study protocol was in accordance with the principles of the Declaration of Helsinki for research involving human participants. Both eyes were examined in all study subjects, and the results from lagophthalmic and contralateral healthy eye of the same patient were compared.

Examinations for the study included a variety of tests that were performed in the following sequence: best-corrected visual acuity (BCVA), tear film osmolarity measurement, tear sample collection for MMP-9 testing, degree of lagophthalmos measurement, sensitivity testing, corneal fluorescein staining, and tear break-up time (TBUT). All measurements and tests were performed in the same room, between 9:00 and 11:00 a.m., with similar temperature and humidity to minimize external factors.

### Clinical testing

Tear film osmolarity was measured using the TearLab Osmolarity System (Tearlab Corp.) according to the manufacturer´s instructions; a 50-nL sample of tear film was obtained from the outer lower tear meniscus. The patients were asked not to apply any artificial tear drops for at least an hour before the examination. The lagophthalmic eye was measured first. The measurement range of the system is 270–400 mOsm/L [9, 10]. If values were below the

measurement range, a value of 270 mOsm/L was recorded. The degree of lagophthalmos was measured in mm using a calliper while attempting eyelid closure. Mechanical corneal sensitivity was tested using a Cochet-Bonnet esthesiometer (Luneau Ophthalmologie, Chartres, France). The esthesiometer has a 0.12 mm-diameter nylon filament with a length in the range of 0.5–6.0 cm; longer filament lengths generated lower filament pressures. A positive response was defined as the patient felt the filament was touching the cornea. The longest filament length resulting in a positive response in three consecutive touches was considered the sensitivity threshold and recorded as the filament length [11]. For TBUT, the tear film was stained with fluorescein sodium, which was instilled into the inferior tear meniscus using a fluorescein sodium ophthalmic strip (I-DEW FLO, Jodimed, Great Britain). The average time to the first break in the tear film was calculated for three consecutive measurements using cobalt blue illumination and a stopwatch. After that, the intensity of ocular surface fluorescein staining was evaluated, and corneal and conjunctival staining levels were graded according to the Oxford Scheme for Ocular Surface Fluorescein Staining (a 0–5 scale in each of the evaluated areas, i.e., medial conjunctiva, cornea, and temporal conjunctiva) [12, 13].

## MMP-9 determination

For MMP-9 evaluation, a non-stimulated tear fluid sample was collected atraumatically from the lower tear meniscus using a glass micropipette. Typically, 1–3 μl of tears were obtained. Each sample was then placed in a 0.5 ml Eppendorf tube and stored at −20˚C until used. The level of MMP-9 was measured using a commercial Human MMP-9 ELISA kit (ThermoFisher Scientific, Human MMP-9 Platinum ELISA). The assay was performed according to the instructions of the manufacturer.

**Statistical analysis.** The obtained values from lagophthalmic and contralateral healthy eye of the same patient were compared. Data are presented as mean ± standard error of the mean (SEM). The nonparametric Wilcoxon signed-rank test was used for statistical analysis of osmolarity, TBUT values, ocular surface fluorescein staining, corneal sensitivity, and tear MMP-9 data. The Mann-Whitney U test was used to compare the tear MMP-9 value in lagophthalmic eyes with and without tarsorrhaphy.

The Spearman rank correlation test was used to test for possible correlations between tear MMP-9 levels and tear osmolarity, surface fluorescein staining, TBUT, and corneal sensitivity for both lagophthalmic and healthy eyes. A possible correlation between the tear MMP-9 levels and the degree of lagophthalmos, and time from the surgery was also tested. StatView 5.0 (SAS Institute Inc., Cary, NC) statistical software was used for the analyses. P values < 0.05 were considered statistically significant.

## Results

Seven males and nine females were included in the study. All results are presented as mean + SEM. Patients age was 53.5 ± 3.5 years. The time from the surgery was 22.1 ± 5,0 months. The degree of lagophthalmos was 3.5 ± 0.6 mm. Nine of the patients had a history of partial external tarsorrhaphy. Five patients had a history of facial nerve/sublingual nerve anastomosis. Detailed characteristics of the cohort are presented in Tables 1 and 2.

TBUT was statistically significantly lower in lagophthalmic eyes (5.5 ± 1.1 s) than in healthy eyes (9.8 ± 0.8 s; p = 0.0009). The Oxford score was higher in lagophthalmic eyes (2.6 ± 0,5) than in healthy eyes (0.0 ± 0.0; p = 0.0010). Statistically, the sensitivity was significantly lower in lagophthalmic eyes (5.6 ± 0.2 mm) than in healthy eyes (6.0 ± 0.0 mm; p = 0.0277). Tear film osmolarity was below the range of lagophthalmic eyes in one patient (No. 13), and in this case, the value was recorded as 270 mOsm/l. There was no statistically significant difference

**Table 1. Cohort characteristics.**

| No | Sex | Age (years) | Time from the primary surgery (months) | BCVA lagophthalmic eye | BCVA healthy eye | Lagophthalmos (mm) | Tarsorrhaphy | Anastomosis |
|----|-----|-----|-----|-----|-----|-----|-----|-----|
| 1 | F | 75 | 81 | 0.7 | 1.0 | 4 | - | - |
| 2 | M | 57 | 48 | 0.8 | 0.9 | 1 | - | - |
| 3 | F | 70 | 32 | 1.0 | 1.0 | 1 | - | - |
| 4 | F | 67 | 16 | (amblyopia) 0.17 | 1.0 | 2 | + | + |
| 5 | F | 47 | 9 | 0.9 | 1.0 | 7 | + | - |
| 6 | F | 57 | 33 | 1.0 | 1.0 | 0 | - | - |
| 7 | F | 47 | 17 | 1.0 | 1.0 | 1 | + | + |
| 8 | F | 69 | 21 | 0.9 | 1.0 | 6 | - | - |
| 9 | M | 44 | 14 | 1.0 | 1.0 | 3 | + | + |
| 10 | M | 51 | 26 | 1.0 | 1.0 | 8 | + | - |
| 11 | F | 67 | 8 | 1.0 | 1.0 | 4 | - | - |
| 12 | F | 48 | 12 | 0.3 | 1.0 | 7 | + | - |
| 13 | M | 20 | 4 | 0.9 | 1.0 | 3 | + | + |
| 14 | M | 47 | 3 | 0.7 | 1.0 | 4 | + | - |
| 15 | M | 47 | 3 | 1.0 | 1.0 | 2 | - | - |
| 16 | M | 43 | 26 | 0.7 | 1.0 | 2 | + | + |

F—female, M—male, BCVA—best corrected visual acuity

**Table 2. Detailed test results.**

| No | MMP-9 lagophthalmic eye (ng/ml) | MMP-9 healthy eye (ng/ml) | osmolarity lagophthalmic eye (mOsm/l) | osmolarity healthy (mOsm/l) | TBUT lagophthalmic eye (s) | TBUT healthy eye (s) | Oxford score lagophthalmic eye | Oxford score healthy eye | Corneal sensitivity lagophthalmic eye | Corneal sensitivity healthy eye |
|----|-----|-----|-----|-----|-----|-----|-----|-----|-----|-----|
| 1 | 316.22 | 637.35 | 327 | 318 | 8 | 13 | 1-2-1 | 0-0-0 | 6 | 6 |
| 2 | 1143.39 | 240.28 | 305 | 294 | 4 | 7 | 0-2-0 | 0-0-0 | 6 | 6 |
| 3 | 61.22 | 24.44 | 301 | 302 | 9 | 12 | 1-0-1 | 0-0-0 | 5.5 | 6 |
| 4 | 180.10 | 10.05 | 285 | 294 | 2 | 10 | 1-1-0 | 0-0-0 | 4.5 | 6 |
| 5 | 186.19 | 53.10 | 304 | 306 | 2 | 8 | 0-0-1 | 0-0-0 | 5.5 | 6 |
| 6 | 173.97 | 150.47 | 293 | 293 | 16 | 18 | 0-0-0 | 0-0-0 | 6 | 6 |
| 7 | 223.62 | 16.10 | 299 | 296 | 14 | 12 | 0-1-0 | 0-0-0 | 6 | 6 |
| 8 | 4328.37 | 209.38 | 320 | 311 | 2 | 5 | 1-2-1 | 0-0-0 | 5.5 | 6 |
| 9 | 2633.11 | 619.50 | 294 | 280 | 8 | 12 | 0-0-0 | 0-0-0 | 6 | 6 |
| 10 | 1723.69 | 116.22 | 285 | 305 | 7 | 6 | 2-2-1 | 0-0-0 | 5.5 | 6 |
| 11 | 108.61 | 268.74 | 286 | 289 | 3 | 8 | 0-1-0 | 0-0-0 | N/A | N/A |
| 12 | 337.01 | 24.19 | 323 | 338 | 2 | 7 | 0-1-1 | 0-0-0 | N/A | N/A |
| 13 | 34353.35 | 77.64 | 270 | 284 | 3 | 12 | 1-2-0 | 0-0-0 | 6 | 6 |
| 14 | 5805.24 | 148.89 | 304 | 299 | 2 | 8 | 2-3-2 | 0-0-0 | 6 | 6 |
| 15 | 2698.62 | 1014.16 | 290 | 297 | 3 | 8 | 2-2-2 | 0-0-0 | 6 | 6 |
| 16 | 1006.18 | 650.12 | 374 | 290 | 3 | 10 | 1-1-0 | 0-0-0 | 4 | 6 |
| | p = 0.0064 | | p = 0.9547 | | P = 0.0009 | | p = 0.0010 | | p = 0.0277 | |

MMP-9—matrix metalloproteinase 9, TBUT—tear break-up time

between osmolarity in lagophthalmic eyes (303.8 ± 6.0 mOsm/L) and healthy eyes (299.8 ± 3.5 mOsm/L; p = 0.9547). Tear film MMP-9 levels varied considerably, and the largest outlier (No. 13) was excluded from our statistical analyses. Tear MMP-9 values were significantly higher in lagophthalmic eyes (1395.0 ± 453.2 ng/ml) than in healthy eyes (278.9 ± 79.0 ng/ml; p = 0.0064). There was no difference between MMP-9 values in lagophthalmic eyes with tarsorrhaphy (1511.9 ± 688.7 ng/ml) and without tarsorrhaphy (1261.6 ± 623.6 ng/ml; p = 0.4875).

Higher MMP-9 values were associated with increased ocular surface fluorescein staining (r = 0.512, p = 0.0022) and lower TBUT (−0.382, p = 0.0419). No association was found between MMP-9 values and osmolarity (r = −0.039, p = 0.8286), and corneal sensitivity (r = −0.148, p = 0.5103). There was no association between MMP-9 values in lagophthalmic eyes and the time from surgery (r = −0.258, p = 0.3393), or the degree of lagophthalmos (r = 0.329, p = 0.2031).

## Discussion

Facial nerve palsy, whether idiopathic or iatrogenic, traumatic, or in malignancy, places the ocular surface at risk. Reduction or absence of m. orbicularis oculi function results in lagophthalmos and corneal exposure, which can also be exacerbated by eyelid malposition. In lagophthalmos resulting from cerebellopontine angle masses a significant decrease in tear production can also stem from lacrimal nerve damage. The disability may be temporary, but management of exposure keratopathy is paramount to prevent corneal breakdown, stromal changes, scarring, and permanent vision loss. Significant exposure keratopathy can be complicated by a loss of corneal sensation, leading to neurotrophic corneal ulcers. However, the duration of facial nerve palsy does not necessarily impact the final BCVA and degree of keratopathy. Initial management consists of artificial tear drops and ointment for corneal lubrication and strategies to approach the lagophthalmos [14, 15].

Tear film instability, which includes both rapid thinning and tear break-up, is considered a core dry eye ocular damage mechanism along with tear film hyperosmolarity. It is thought that tear film instability occurs due to increased evaporation, which leads to increased tear film osmolarity, thereby stressing the ocular surface and leading to a vicious cycle of inflammation and hyperalgesia [3, 16, 17]. A similar state may occur in lagophthalmos due to lid malfunction and increased evaporation.

Traditionally, ocular surface staining and tear film stability testing are assessed using fluorescein surface staining and fluorescein TBUT to look for signs of dry eye [12, 18]. Per the assumptions of our study, we demonstrated an increase in ocular surface staining and decreased tear film stability in lagophthalmic eyes. We also showed a decrease of corneal sensitivity in lagophthalmic eyes [4].

Tear film osmolarity is considered as one of the new markers of dry eye disease [4, 19, 20]. Surprisingly, we did not find abnormal osmolarity values in most of our patients, and tear osmolarity did not differ between lagophthalmic and healthy eyes. However, blink insufficiency may affect tear film stability and tear distribution during the inter-blink period [21]. Thus, the osmolarity of tears sampled from the inferior tear meniscus might undervalue the osmolarity of tear fluid across the cornea in lagophthalmic eyes [2, 22]. Inconsistency in tear film osmolarity measurements, especially in severe dry eye patients, was also described [23–25].

MMP-9 is a proteolytic enzyme that has been implicated in various ocular surface disorders. MMP-9 has also been considered as a possible clinical indicator of ocular surface disease and inflammation [8, 16, 20]. Its production by the corneal epithelium and increase of MMP-9 tear

levels could be affected by tear film hyperosmolarity and desiccating stress exposure [26–28]. MMPs can also disrupt the corneal barrier and promote corneal permeability and irregularities, which could result in corneal ulceration [27, 29]. Furthermore, some studies even suggested that MMP-9 levels might be a more sensitive diagnostic marker than the clinical signs [30–32]. In our study, MMP-9 values were statistically significantly higher in lagophthalmic eyes compared to healthy eyes. We found a correlation between tear MMP-9 levels and fluorescein ocular surface staining as well as TBUT. Surprisingly, MMP-9 levels were also higher than the presumed normal values in healthy eyes, but immune reactions had been previously described in the other healthy eye in cases of unilateral inflammation [33].

The term lagophthalmos refers to a condition associated with incomplete closing of the eye and represents a risk factor of corneal exposure and the development of surface inflammation [4, 5]. The degree of eyelid closure deficit differs between individual patients. It could be assumed that ocular surface injury would be greater with a more pronounced lagophthalmos degree. However, we found no associations between MMP-9 and the degree of lagophthalmos regardless of tarsorrhaphy. A fundamental approach to prevent ocular surface damage is sufficient lubrication. Thus, the MMP-9 may also reflect the compliance of patients with the recommended treatment and may suggest that MMP-9 levels depend more on proper ocular surface lubrication than lagophthalmos severity.

Increased MMP-9 levels upon awakening were described [34]. Environmental factors or systemic inflammatory disease may also influence the MMP-9 tear level [28, 35]. The high variability of tear MMP-9 levels may be a potential problem for its clinical use. In our study, the variance of MMP-9 values was considerable, especially within lagophthalmic eyes. It may result from the different compliance with lubrication therapy, and also from more succeptibilty of the lagophthalmic eye to environmental insults. However, the importance of MMP-9 testing may stem from comparing the value with contralateral healthy eye.

## Conclusions

Lagophthalmos increases the risk of ocular surface exposure and damage. MMP-9 has been considered as a clinical indicator of ocular surface inflammation. Our study suggests, that tear MMP-9 level monitoring could help clinicians control the ocular surface damage in these patients. It may contribute to detecting progression of ocular surface inflammation and facilitate timely changes in treatment. Furthermore, higher tear MMP-9 levels could help in screening and sorting out the patients whom are more likely to profit from treatment with anti-inflammatory medications [31]. However, exact mechanism of dry eye development in different underlying conditions is still under investigation [36], and further studies are required to understand the potential role of tear MMP-9 measurements in lagophthalmic patients.

## Acknowledgments

We would like to thank Azzat Al-Redouan and Thomas Secrest for the help with reviewing the language content.

## Author Contributions

**Conceptualization:** Zdeněk Fík, Gabriela Mahelková.

**Data curation:** Marcela Minaříková, Zdeněk Fík, Josef Štorm, Kateřina Helisová, Květoslava Ferrová.

**Formal analysis:** Josef Štorm, Kateřina Helisová, Gabriela Mahelková.

**Investigation:** Marcela Minaříková, Josef Štorm, Květoslava Ferrová.

**Methodology:** Marcela Minaříková, Zdeněk Fík, Kateřina Helisová, Květoslava Ferrová, Gabriela Mahelková.

**Project administration:** Marcela Minaříková, Josef Štorm, Květoslava Ferrová.

**Resources:** Marcela Minaříková, Zdeněk Fík.

**Software:** Marcela Minaříková, Josef Štorm, Kateřina Helisová.

**Supervision:** Gabriela Mahelková.

**Validation:** Marcela Minaříková, Zdeněk Fík, Kateřina Helisová, Gabriela Mahelková.

**Writing – original draft:** Marcela Minaříková.

**Writing – review & editing:** Zdeněk Fík, Josef Štorm, Gabriela Mahelková.

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
