## [Decision Letter · Decision Letter 0]

14 Jul 2022

PONE-D-22-11297Tear matrix metalloproteinase-9 levels may help to follow a ocular surface injury in lagophthalmic eyesPLOS ONE

Dear Dr. Mahelkova,

Thank you for submitting your manuscript to PLOS ONE. After careful consideration, we feel that it has merit but does not fully meet PLOS ONE’s publication criteria as it currently stands. Therefore, we invite you to submit a revised version of the manuscript that addresses the points raised by the second reviewer.

We look forward to receiving your revised manuscript.

Kind regards,

Deepak Shukla

Academic Editor

PLOS ONE

Journal Requirements:

a) Did participants provide their written or verbal informed consent to participate in this study?

3. Thank you for stating the following in the Acknowledgments Section of your manuscript: "Supported by project No. 1110026 of the Second Faculty of Medicine, Charles University, Prague and by the project (Ministry of Health, Czech Republic) for conceptual development of research organization 00064203 (University Hospital Motol, Prague, Czech Republic).

Financial disclosure: None"

Please remove any funding-related text from the manuscript and let us know how you would like to update your Funding Statement. Currently, your Funding Statement reads as follows: "The author(s) received no specific funding for this work."

Reviewers' comments:

Reviewer's Responses to Questions

**Comments to the Author**

1. Is the manuscript technically sound, and do the data support the conclusions?

Reviewer #1: Partly

Reviewer #2: Yes

2. Has the statistical analysis been performed appropriately and rigorously? 

Reviewer #1: Yes

Reviewer #2: Yes

3. Have the authors made all data underlying the findings in their manuscript fully available?

Reviewer #1: Yes

Reviewer #2: Yes

4. Is the manuscript presented in an intelligible fashion and written in standard English?

Reviewer #1: Yes

Reviewer #2: Yes

5. Review Comments to the Author

Reviewer #1: The manuscript (PONE-D-22-11297) looks technically in shape with some caveats. For example, in lines 234-237, it's not entirely clear how author is concluding 'the MMP-9 may also reflect the compliance of patients with treatment ...' based on the association between MMP-9 and degree of lagophthalmos. A clearer explanation would have been better. The statistical analysis looks okay, though use of paired t-test raises some doubt since the sample size is small 16 and use of parametric test is not justifiable. Wilcoxon signed-rank test would've been a better option. The article could've been structured better by including a section with results/conclusion. Numerous articles have been published previously, which suggest measurement of ocular surface MMP-9 level provides a useful marker for ocular inflammation. Hence, some of the suggestions/ findings made in this manuscript seem repetitive and hence not novel.

The study is a step in the right direction but findings need to evaluated with larger sample sizes.

Reviewer #2: In the manuscript titled “Tear matrix metalloproteinase-9 levels may help to follow a ocular surface injury in lagophthalmic eyes”, the authors have evaluated the possible usefulness of measuring tear film levels of MMP-9 in patients with lagophthalmos and suggest that tear MMP-9 levels correlate with ocular surface damage and will help in timely clinical assessment and changes in treatment regimen.

Here are some minor concerns:

1. The study is well done though the sample size is rather small at n=16 (7 men and 9 women).

2. The manuscript needs to be re-checked for English grammar and sentence construction.

3. Although the authors have explained this in the “Methods” section, the term “Oxford Score” in the “Results” section appears very vague since this score is used for multiple measurements in different disease conditions, Oxford grading score for ocular surface staining, knees, muscles, etc. The authors need to specify throughout the manuscript the term “Oxford Scheme for Ocular Surface Fluorescein Staining” for ease of understanding for the reader.

4. The authors are advised to calculate the “Mean + SEM”. This is recommended since the “Standard Deviation value” for Tear MMP-9 values in lagophthalmic eyes is higher than the “Mean value” (Line 175).

5. The “Standard Deviation value” is almost as high as the Mean value for TBUT as well as “Oxford Score” in lagophthalmic eyes (Line 165 and Line 166)

6. The word “Mean + SEM” should precede the values for the same mentioned in the “Results section”. It is not easy for the reader to know what the numbers in parenthesis actually signify.

7. The authors admit that there is considerable variance in the MMP-9 values within the lagophthalmic group but do not discuss the possible reasons for this. This needs to be discussed in greater detail along with relevant references, which will enhance the quality of the manuscript.

6. PLOS authors have the option to publish the peer review history of their article (what does this mean?). If published, this will include your full peer review and any attached files.

Reviewer #1: No

Reviewer #2: No

---

## [Author Response · Author response to Decision Letter 0]

19 Aug 2022

Dear Reviewer #1,

Thank you for your comments on our manuscript "Tear matrix metalloproteinase-9 levels may help to follow a ocular surface injury in lagophthalmic eyes". 

We have gone through them carefully and revised the manuscript accordingly. 

We supplemented the discussion detailing the effect of patient’s compliance with therapy on the ocular surface (originally lines 234-237). As per the recommendation we have replaced the use of paired t-test with Wilcoxon singed-rank test. We have also included a Conclusions section. 

We agree that many articles have been published already, which suggests that measurement of ocular surface MMP-9 levels can be a useful marker in revealing ocular inflammation. Rather, our study aimed to test the usefulness of this approach in a specific group of patients with lagophthalmos.

Dear Reviewer #2,

Thank you for your comments on our manuscript "Tear matrix metalloproteinase-9 levels may help to follow a ocular surface injury in lagophthalmic eyes". 

We have gone through them carefully and revised the manuscript accordingly.

We have clarified the name of the test – Oxford Scheme for Ocular Surface Fluorescein Staining test.

We replaced the format of the results and present them now as mean ± SEM.

We added the information about the results format to the beginning of Results section.

We have supplemented the discussion on the considerable variance in MMP-9 values. We have also added a reference (No.35).

Thank you for your comments on our manuscript "Tear matrix metalloproteinase-9 levels may help to follow a ocular surface injury in lagophthalmic eyes". 

1. We have gone through them carefully and revised the manuscript accordingly.We verified the manuscript to meet PLOS ONE’s style requirements.

2. We added the information regarding the form of informed consent.

3. We apologize for placing the funding declaration improperly. We removed it from the Acknowledgements section and included the statement in the cover letter. Thank you for informing us and suggesting the correction.

4. We also included the Data Availability statement within the cover letter. The minimal dataset underlying our results described in our manuscript is presented within Tables 1 and 2 of the manuscript.

5. We reviewed the manuscript reference list. According to Reviewer 2 comments we added a reference (No. 35).

---

## [Editor Report · Decision Letter 1]

24 Aug 2022

Tear matrix metalloproteinase-9 levels may help to follow a ocular surface injury in lagophthalmic eyes

PONE-D-22-11297R1

Dear Dr. Mahelkova,

We’re pleased to inform you that your manuscript has been judged scientifically suitable for publication and will be formally accepted for publication once it meets all outstanding technical requirements.

Kind regards,

Deepak Shukla

Academic Editor

PLOS ONE
---

## [Editor Report · Acceptance letter]

1 Sep 2022

PONE-D-22-11297R1 

Tear matrix metalloproteinase-9 levels may help to follow a ocular surface injury in lagophthalmic eyes 

Dear Dr. Mahelková:

I'm pleased to inform you that your manuscript has been deemed suitable for publication in PLOS ONE. Congratulations! Your manuscript is now with our production department. 

Kind regards, 

on behalf of

Prof. Deepak Shukla 

Academic Editor

PLOS ONE